# Intravoxel Incoherent Motion Diffusion-Weighted MRI, Fat Quantification, and Electromyography: Correlation in Polymyositis and Dermatomyositis

Hyunjung Kim [1,*], Sang Yeol Yong [2], Chuluunbaatar Otgonbaatar [3] and Seoung Wan Nam [4]

1   Department of Radiology, Wonju Severance Christian Hospital, Wonju College of Medicine,
    Yonsei University of Korea, Wonju 26426, Republic of Korea
2   Department of Rehabilitation Medicine, Wonju Severance Christian Hospital, Wonju College of Medicine,
    Yonsei University of Korea, Wonju 26426, Republic of Korea; rehsyyong@yonsei.ac.kr
3   Department of Radiology, College of Medicine, Seoul National University, Seoul 03080, Republic of Korea;
    chukarad@gmail.com
4   Department of Rheumatology, Wonju Severance Christian Hospital, Wonju College of Medicine,
    Yonsei University of Korea, Wonju 26426, Republic of Korea; namsw@yonsei.ac.kr
*   Correspondence: radkhj@yonsei.ac.kr; Tel.: +82-33-742-1472

**Abstract:** (1) Background: The intravoxel incoherent motion (IVIM) model can provide information about both molecular diffusion and blood flow for the evaluation of skeletal muscle inflammation. MRI-based fat quantification is advantageous for assessing fat infiltration in skeletal muscle. (2) Purpose: We aimed to quantitatively measure various parameters associated with IVIM diffusion-weighted imaging (DWI) and fat quantification in the muscles of patients with polymyositis and dermatomyositis using magnetic resonance imaging and to investigate the relationship between these parameters and electromyography (EMG) findings. (3) Material and methods: Data were retrospectively evaluated for 12 patients with polymyositis and dermatomyositis who underwent thigh MRI, including IVIM-DWI and fat quantification. The IVIM-derived parameters included the pure diffusion coefficient (D), pseudodiffusion coefficient (D*), and perfusion fraction ($f$). Fat fraction values were assessed using the six-point Dixon technique. Needle EMG was performed within 9 days of the MRI. (4) Results: The $f$ values (19.02 ± 4.87%) in muscles with pathological spontaneous activity on EMG were significantly higher than those (14.60 ± 5.31) in muscles without pathological spontaneous activity ($p < 0.027$). There were no significant differences in D, D*, ADC, or fat fraction between muscles with and without pathologic spontaneous activity. Significant negative correlations were observed between fat fraction and amplitude ($r = -0.402$, $p < 0.015$) and between fat fraction and duration ($r = -0.360$, $p < 0.031$). (5) Conclusion: The current study demonstrates that IVIM-DWI and fat quantification using 3.0 T MRI may aid in predicting EMG findings in patients with polymyositis and dermatomyositis and promote the pathophysiological study of idiopathic inflammatory myopathies.

**Keywords:** electromyography; fat fraction; idiopathic inflammatory myopathy; intravoxel incoherent motion diffusion-weighted imaging; magnetic resonance imaging

## 1. Introduction

Idiopathic inflammatory myopathies (IIM) include chronic systemic connective tissue diseases such as dermatomyositis (DM), polymyositis (PM), inclusion body myositis, necrotizing autoimmune myositis, and overlap myositis [1]. Diagnoses of PM and DM, two classical forms of IIM [2], are made based on a combination of clinical features and findings from laboratory tests, needle electromyography (EMG), and muscle biopsy. Although Bohan and Peter proposed the diagnostic criteria for PM and DM in 1975 [1,2], magnetic resonance imaging (MRI) has recently been considered a valuable modality for evaluating

myopathies, guiding muscle biopsies, and monitoring disease progression in patients with myopathy [3–5].

Intravoxel incoherent motion (IVIM) diffusion-weighted imaging (DWI) is a model to separate free water motion and incoherent motion components from multiple data sets of DWI with different *b* values [6]. This method has been more accessible due to the advantages of the MRI technique and gradient strength. The IVIM model is sensitive not only to the extravascular molecular diffusion of water but also to the intravascular motion of blood flow in capillaries using sufficiently low multiple *b*-values [6,7]. The IVIM model has been applied to highly vascular tissues and can provide information about both molecular diffusion within the tissue and blood flow [7,8], making it an attractive tool for the evaluation of skeletal muscle injury caused by inflammatory conditions, such as PM or DM, without a contrast agent.

MRI-based fat quantification is advantageous for assessing fat infiltration in skeletal muscle, which often occurs in cases of chronic inflammation. Since the introduction of chemical-shift imaging (CSI) by Dixon in 1984, this technique has revealed inherent differences in the resonant frequencies between lipids and water [9], allowing for the calculation of the fat fraction based on the different signals from water and fat [10]. Research has also demonstrated a strong correlation between data acquired using two- and three-point Dixon sequences and the ratio of fat determined via muscle biopsy [11].

Needle electromyography (EMG), an important diagnostic modality in clinical practice, has been used to differentiate other neurogenic disorders and monitor disease activity in patients with myopathy [1]. Patients with IIM often exhibit small amplitudes, short durations, or polyphasic potentials on EMG [12]. In addition, pathological spontaneous activities such as fibrillation potentials and positive sharp waves (Fib/PSWs) are associated with muscle fiber degeneration and active myositis [12,13].

However, despite the above findings, it remains unclear whether there is a correlation between quantitative MRI and EMG findings in patients with PM and DM. Therefore, in the present study, we aimed to quantitatively measure various parameters associated with IVIM-DWI and fat quantification in the muscles of patients with PM and DM using MRI and to investigate the relationship between these parameters and EMG findings.

## 2. Materials and Methods

The appropriate institutional review board approved this retrospective study and waived the requirement for informed consent (CR321157).

### 2.1. Patients

We retrospectively analyzed data for all patients with histopathologically confirmed PM or DM identified between December 2020 and December 2021. The clinical features of the patients are presented in Table 1. The study group comprised four men and eight women with a mean (±11.82) age of 51.3 (range: 35–76 years). All patients underwent MRI and EMG. The time interval between MRI and EMG ranged from 1 to 9 days (mean: 5.7 days). Patients who had received steroid therapy before MRI and EMG were excluded from the study.

### 2.2. MRI Protocols

In all cases, MRI of the bilateral thighs was performed using a 3.0 T scanner (Magnetom Vida, Siemens Healthcare, Erlangen, Germany). Standard MR images were obtained in accordance with our routine clinical protocol, including axial and coronal T1-weighted turbo spin-echo (TSE) images, axial and coronal T2 Dixon TSE images with in-phase and water-only images, and axial and coronal fat-suppressed contrast-enhanced T1-weighted TSE images.

IVIM-DWI was performed prior to the administration of a contrast agent using single-shot spin-echo echo-planar imaging (EPI) with monopolar diffusion gradients. A parallel imaging technique (simultaneous multi-slice imaging) with a two-fold acceleration factor

was used to shorten the echo train length. Encoding was performed in three orthogonal directions with a series of eleven *b*-values (0, 20, 50, 80, 100, 150, 200, 500, 800, and 1500 s/mm$^2$). The acquisition time was 293 s. The scanning parameters were as follows: matrix = 140 × 140; field of view (FOV) = 400 × 400 mm; repetition time (TR) = 2900 ms; echo time (TE) = 68 ms; slice thickness = 8 mm; and EPI factor = 140. The apparent diffusion coefficient (ADC) was calculated on a voxel-by-voxel basis using a biexponential non-linear fit using the equation below [14]. The pseudodiffusion coefficient (D*), pure diffusion coefficient (D), and perfusion fraction (*f*) were calculated using a biexponential IVIM model with multiple *b* values.

$$\frac{S(b)}{S_0} = (1 - f) \times e^{-bD} + f \times e^{-bD^*}$$

**Table 1.** Clinical Characteristics of Patients.

| No. | Sex | Age | Diagnosis | Disease Duration (Months) * | Time Interval between MRI and EMG (Days) | Fib | PSW |
|-----|-----|-----|-----------|------------------------------|-------------------------------------------|-----|-----|
| 1 | M | 47 | DM | 1 | 1 | + | + |
| 2 | F | 55 | DM | 1 | 4 | + | + |
| 3 | F | 61 | DM | 2 | 9 | - | - |
| 4 | F | 39 | DM | 1 | 7 | + | - |
| 5 | M | 38 | DM | 2 | 9 | - | - |
| 6 | F | 56 | DM | 24 | 2 | + | + |
| 7 | F | 76 | DM | 12 | 7 | + | + |
| 8 | F | 35 | PM | 10 | 1 | + | + |
| 9 | F | 56 | DM | 3 | 5 | - | - |
| 10 | F | 51 | PM | 2 | 6 | + | + |
| 11 | M | 60 | PM | 1 | 8 | + | + |
| 12 | M | 42 | PM | 9 | 9 | + | + |

Note: * Disease duration before MRI and EMG was determined based on the patient's clinical history. DM: dermatomyositis; EMG: electromyography; Fib: fibrillation potentials; MRI: magnetic resonance imaging; PM: polymyositis; PSW: positive sharp waves.

Post-processing of the acquired DWI data was performed using prototype software provided by the manufacturer (Syngo Via Frontier MR Body Diffusion Toolbox, version V1.6.0; Siemens Healthcare, Erlangen, Germany).

The six-point Dixon technique was used for fat quantification. Volumetric interpolated breath-hold examination (VIBE) gradient echo sequences with T2*-correction images were acquired in the axial plane using controlled aliasing in parallel imaging results in higher acceleration (CAIPIRINHA). The scanning parameters were as follows: matrix = 160 × 111; FOV = 400 × 350 mm; TR = 9 ms; TE = 1.05, 2.46, 3.69, 4.92, 6.15, and 7.38 ms; slice thickness = 4 mm; flip angle = 4°. The acquisition time was 26 s. The fat fraction maps, water fraction maps, and R2* maps were generated automatically by a post-processing algorithm based on a multi-step adaptive fitting method [15] using a multiple-peak fat model.

The scan parameters of IVIM-DWI and the six-point Dixon technique are summarized in Table 2.

*2.3. Image Analysis*

MR images were retrospectively analyzed by one general radiologist (with thirteen years of experience) and one musculoskeletal radiologist (with five years of experience) independently. IVIM-derived parameters, ADC values, and fat fractions were retrospectively measured at three sites in the muscle where EMG was performed. The one muscle per patient was analyzed based on medical records and images. A region of interest (ROI) spanning 100 mm$^2$ was manually drawn within the muscles exhibiting high signal intensities on high *b*-value DWI. Fat quantification was performed at the same sites in the same muscle, with an ROI of 100 mm$^2$ using a fat-fraction map. Both ROIs were placed in a single

slice. IVIM-derived parameters and ADC values were measured using prototype software provided by the manufacturer (Syngo Via Frontier MR Body Diffusion Toolbox, version V1.6.0; Siemens Healthcare). Fat fraction measurements were performed using commercially available software (Syngo Via, version VB30A, Siemens Healthcare). A representative example of ROI placement is shown in Figure 1.

**Table 2.** MRI Parameters for IVIM-DWI and six-point Dixon technique.

| Parameter | IVIM-DWI | T1 VIBE Multi-Echo DIXON |
|---|---|---|
| FOV (mm) | $400 \times 400$ | $400 \times 350$ |
| Matrix size | $140 \times 140$ | $160 \times 111$ |
| TR (ms) | 2900 | 9 |
| TE (ms) | 68 | 1.05, 2.46, 3.69, 4.92, 6.15, 7.38 |
| Flip angle ° | - | 4 |
| Fat suppression | SPAIR | |
| Slice thickness (mm) | 8 | 4 |
| Intersection Gap (mm) | 0.8 | 0 |
| EPI factor | 140 | - |
| Number of excitations | 3 | - |
| Acquisition time (s) | 293 | 26 |

Note: DWI: diffusion-weighted imaging; EPI: echo-planar imaging; FOV: field of view; IVIM: intravoxel incoherent motion; TE: echo time; TR: repetition time; VIBE: volumetric interpolated breath-hold examination.

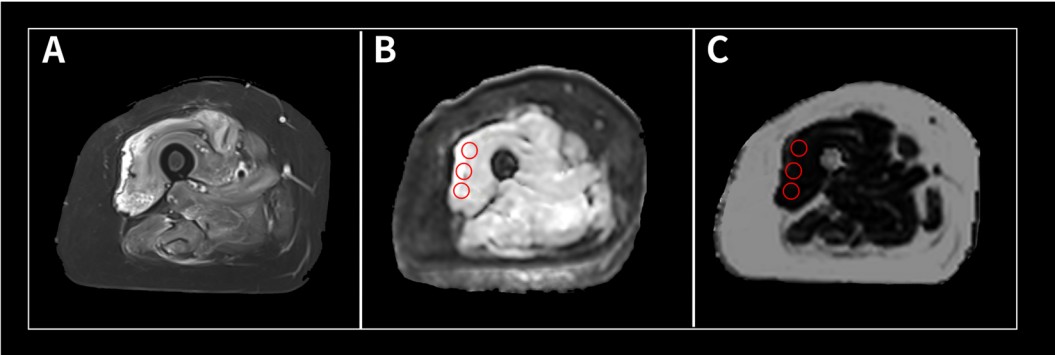

**Figure 1.** Image processing in a 51-year-old female patient with polymyositis. (**A**) Axial T2-weighted Dixon water image showing multifocal high signal intensities in the right thigh muscles, especially the vastus lateralis muscle. (**B**) Axial IVIM-DWI. The ROIs (circle) were placed within the vasus lateralis muscle. (**C**) Fat fraction map. The ROIs (circle) were placed within the site corresponding to the ROI on IVIM-DWI. IVIM-DWI: intravoxel incoherent motion diffusion-weighted image; ROI: region of interest.

## 2.4. Electromyography

In accordance with the routine procedure of our electrodiagnostic laboratory in the Department of Rehabilitation Medicine, an EMG study was performed after motor and sensory nerve conduction. Abnormal pathological spontaneous activity was measured. Prior to EMG, the radiologist marked at the same site exhibiting high echogenecities on ultrasound and high signal intensities on the MRI fat-saturated T2-weighted image. The voluntary motor unit action potential (MUAP) was obtained for each examined muscle such that at least 20 MUAPs of simple shape were recorded. Duration and peak-to-peak amplitude were determined for the identified MUAPs. Pathological spontaneous activities such as fibrillation potentials and positive sharp waves (Fib/PSWs) were evaluated.

## 2.5. Statistical Analysis

Statistical analysis was performed using SPSS software (version 25.0; IBM Corp., Armonk, NY, USA). Continuous variables are presented as the mean $\pm$ standard deviation.

Shapiro–Wilk and Kolmogorov–Smirnov tests were used to evaluate normality. An interobserver agreement analysis was performed using the Cohen κ coefficient, whereby a κ value of more than 0.81 was perfect, between 0.61 and 0.8 was substantial, between 0.41 and 0.6 was moderate, and less than 0.4 was fair. The IVIM-derived parameters were compared using an unpaired *t*-test between muscles with pathologic spontaneous activities and those without. Pearson's correlation coefficients (r) were used to assess the correlations among IVIM-derived parameters (D*, D, and *f*), ADC value, fat fraction, and EMG-derived parameters (duration and amplitude). In the correlation analyses, r values were categorized as follows: >0.8, very strong; 0.60–0.79, strong; 0.40–0.59, moderate; 0.20–0.39, weak; 0.00–0.19, very weak. Statistical significance was set at a *p*-value of <0.05.

### 3. Results

Twelve patients (mean age: 51 ± 11.82 years, range: 35–76 years) were included in the analysis. The overall interreader agreements for the quantitative analysis of IVIM-DWI images and fat fraction were excellent (κ > 0.937, Table 3).

**Table 3.** Results of interobserver agreement analysis.

| Parameters | κ (Kappa) | 95% CI |
|---|---|---|
| ADC ($\mu$m$^2$/s) | 0.937 | 0.864–0.969 |
| D ($\mu$m$^2$/s) | 0.951 | 0.899–0.976 |
| D* ($\mu$m$^2$/s) | 0.943 | 0.884–0.972 |
| *f* (%) | 0.958 | 0.918–0.979 |
| Fat (%) | 0.978 | 0.956–0.989 |

Table 4 summarizes the IVIM-derived parameters, fat fraction, and ADC values among muscles with or without pathological spontaneous activity. The *f* values were significantly higher in muscles with pathological spontaneous activity (19.02 ± 4.87) than in those without (14.60 ± 5.31) ($p < 0.027$). D* values ($p = 0.147$), D ($p = 0.724$), fat fractions ($p = 0.160$), and ADC values ($p = 0.747$) were comparable between muscles with and without pathological spontaneous activity. The mean amplitude was 468.33 ± 370.85 mV, whereas the mean duration was 7.89 ± 4.06 ms. There was a significant moderate negative correlation between amplitude and fat fraction (r = −0.402, $p < 0.015$). In addition, a weak yet significant negative correlation was observed between duration and fat fraction (r = −0.360, $p < 0.031$) (Figures 2 and 3).

**Table 4.** Analysis of IVIM-Derived Parameters, Fat Fraction, and ADC Value.

| Parameters | Pathologic Spontaneous Activity | No Pathologic Spontaneous Activity | *p* Value |
|---|---|---|---|
| D* ($\mu$m$^2$/s) | 19,285.18 ± 5615.90 | 15,867.78 ± 7061.79 | 0.147 |
| D ($\mu$m$^2$/s) | 1513.44 ± 289.14 | 1553.02 ± 289.00 | 0.724 |
| *f* (%) | 19.0 ± 4.9 | 14.6 ± 5.3 | 0.027 |
| ADC ($\mu$m$^2$/s) | 1515.97 ± 255.62 | 1547.81 ± 250.80 | 0.747 |
| fat (%) | 8.35 ± 10.27 | 3.37 ± 1.69 | 0.160 |

Note: ADC: apparent diffusion coefficient; D*: pseudodiffusion coefficient; D: pure diffusion coefficient; *f*: perfusion fraction; IVIM: intravoxel incoherent motion.

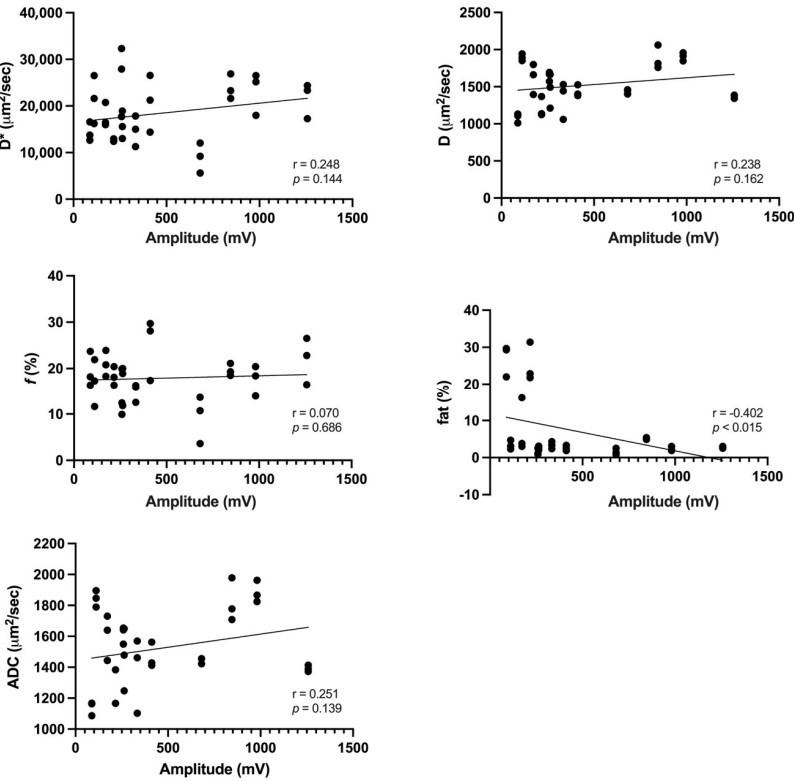

**Figure 2.** Correlations of IVIM-derived parameters, fat fraction, and ADC value with EMG amplitude. There was a significant moderate negative correlation between amplitude and fat fraction ($r = -0.402$, $p < 0.015$). ADC, apparent diffusion coefficient.

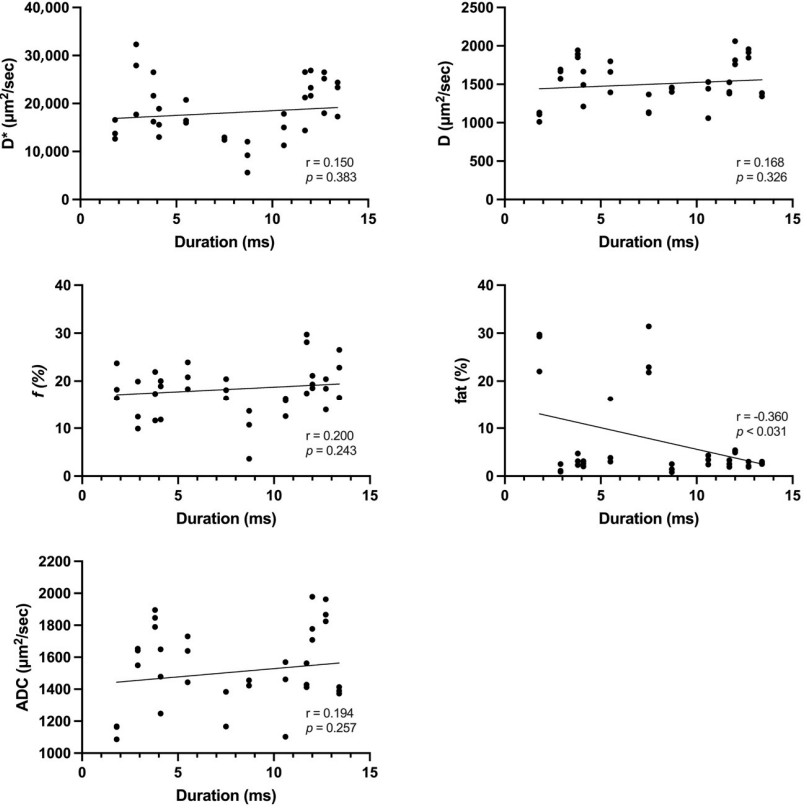

**Figure 3.** Correlations of IVIM-derived parameters, fat fraction, and ADC value with EMG duration. We observed a weak yet significant negative correlation between duration and fat fraction.

## 4. Discussion

To the best of our knowledge, this is the first study to investigate the correlation of EMG findings with IVIM-DWI parameters and fat fraction in patients with PM and DM. Our study demonstrated a relationship between EMG findings and muscle perfusion using IVIM-DWI. Furthermore, we observed a correlation between EMG findings and quantification of fatty degeneration in the affected muscles caused by chronic inflammation. The IVIM-derived perfusion fraction (*f*) was significantly higher in muscles exhibiting pathological spontaneous activity on EMG. However, no significant differences in fat fraction or other IVIM-derived parameters, such as D, D*, and ADC, were observed between the two groups. The fat fraction exhibited a negative correlation with the amplitude and duration of MUAPs.

Complex tissue changes are known to occur in patients with IIM. T-cells invade the endomysium of affected muscles, leading to inflammatory reactions that induce apoptosis and necrosis of muscle fibers [1,16]. Affected muscles exhibit increased accumulation of intracellular and extracellular fluid, cellular and vascular permeability, and membrane disruption [17,18]. Fibrillation and positive waves represent two forms of pathologic spontaneous activity on EMG that can be detected in IIM involving significant muscle necrosis or inflammation. These phenomena are caused by the denervation of some fibers due to muscle fiber necrosis [19]. In our study, *f* values were 23% higher in muscles with pathologic spontaneous activity on EMG than in those without pathologic spontaneous activity. IVIM-DWI is a unique noninvasive tool for investigating microvascular perfusion in muscles [20–22]. The perfusion fraction (*f*) values obtained via IVIM-DWI represent the relative contribution of intravascular signal sources to extravascular signal sources [7], as well as the fractional volume of capillary blood flow in each voxel [23]. A larger *f* value in inflamed muscles may reflect an increased capillary bed and may be related to increased capillary transport of oxygen and metabolites [24]. Increased capillary bed and blood flow due to inflammation may cause denervation of muscle fibers with significant inflammation or necrosis. We observed no significant difference in ADC values between muscles with pathologic spontaneous activities and those without, consistent with the report by Meyer et al. [12].

In a prior study, Ai et al. reported that in patients with polymyositis (PM) and dermatomyositis (DM), the apparent diffusion coefficient (ADC) value in edematous muscles is higher compared to normal and unaffected muscles. Our result is slightly lower than the value reported by Ai et al. [25]. These differences in ADC values may result from the difference in imaging parameters and the anatomic difference in the measured muscle. The increase in the diffusivities of water molecules in edematous muscles reflects active inflammation [25]. However, our study did not show a significant relationship between the apparent diffusion coefficient (ADC) and the electromyography (EMG) findings.

In a prior study using intravoxel incoherent motion diffusion-weighted imaging (IVIM-DWI MRI) on patients with DM, Sigmund et al. reported that the pseudodiffusion coefficient (D*) of quadriceps muscle was decreased compared to the healthy group. The slowdown in pseudodiffusivities may reflect capillary depletion. The pseudodiffusion coefficient (D*) reported in this study was slightly lower than the values measured in the quadriceps muscle by Sigmund et al. [26]. This difference is thought to be due to the ROIs in our study being in a part of the muscle where inflammation was visually confirmed, whereas the prior study used segmentation of each quadriceps muscle. The capillary depletion in areas with visible inflammation would be more severe than in areas without visible inflammation.

Fatty cell infiltration is a consequence of chronic inflammation. Fat deposition within and between muscle fibers makes the affected muscle tissue less organized and reduces the overall tissue anisotropy [7,24]. Low-amplitude and short-duration MUAPs are characteristic EMG features observed in patients with inflammatory myopathies [27]. In the current study, we observed a negative correlation between fat fraction and both the duration and amplitude of MUAPs, suggesting that loss and disorganization of muscle fibers due to fat

deposition leads to decreases in MUAP parameters. Meyer et al. [12] identified a strong positive correlation between MUAP duration and ADC values. In contrast, our study revealed no statistically significant correlations between MUAP duration and ADC values. Ran et al. [28] reported that ADC values were lower in affected muscles than in control muscles, while Meyer et al. [12] hypothesized that a smaller ADC value indicates more severe myositis. As our study included patients with relatively early-stage inflammatory myopathies, this may explain the lack of a significant correlation between ADC and MUAP duration.

In our study, two of the participants showed a fat fraction of more than 20% in the affected muscles, whereas the remaining ten patients exhibited a fat fraction of less than 7%. Due to the small number of patients with a high fat fraction, the results in terms of duration and amplitude on their EMG could appear more pronounced comparatively. These patients had disease durations of 10 and 24 months, respectively, which are relatively longer compared to other patients. This suggests that future studies may need to include a larger group of patients with a variety of disease durations and higher fat fractions for further validation. All patients with a fat fraction of more than 20% showed pathologic spontaneous activities on EMG. The fat fraction serves as a quantitative indicator of fatty cell infiltration in the affected muscle. Our study confirmed the presence of pathologic spontaneous activities on EMG, which can result from severe fatty cell infiltration due to chronic inflammation, leading to the loss and disorganization of muscle fibers.

Our study had several limitations. Our study was a retrospective study, and there is a possibility of selection bias. To avoid selection bias, we recruited consecutive patients who met the inclusion criteria. In our study, a healthy population as a control group was not included. Among them, the study population was small owing to the rarity of this disorder, and only PM and DM were assessed. The study population consisted of patients with early inflammatory myopathies who had a disease duration of less than one year, except for one person, which made it impossible to observe findings at various stages. Further, the ROIs were placed only within a single slice, therefore pathological changes including EMG findings may not have been fully represented.

Our study is the first to compare IVIM-DWI parameters and fat fraction, which are quantitative MRI measures, with EMG findings in inflammatory myopathy. Barsotti et al. reported that in patients with PM and DM, edema evaluated through thigh MRI is associated with disease activity, laboratory findings such as creatine kinase, and physical examination such as manual muscle testing [29]. Additionally, our study revealed that the fat fraction and intravoxel incoherent motion (IVIM) parameters measured by MRI are associated with the electromyography (EMG) findings. EMG is an important tool for predicting degeneration and pathologic changes in affected muscles in the diagnosis of inflammatory myopathy. However, EMG is an invasive diagnostic tool and cannot be used if the patient's condition is unstable. Predicting EMG findings using MRI could offer an alternative for patients where EMG cannot be performed. Moreover, quantitative analysis using MRI can enhance the understanding of the pathophysiology of inflammatory myopathy and enable monitoring of the disease's treatment progress.

## 5. Conclusions

In conclusion, the current study demonstrates that IVIM-DWI and fat quantification using 3.0 T MRI may aid in predicting EMG findings in patients with IIM and promote the pathophysiological study of inflammatory myopathies.

**Author Contributions:** Conceptualization, H.K. and S.Y.Y.; methodology, H.K. and S.Y.Y.; software, H.K. and C.O.; validation, H.K. and S.Y.Y.; formal analysis, H.K. and S.Y.Y.; investigation, H.K., S.W.N. and S.Y.Y.; resources, H.K.; data curation, H.K. and S.Y.Y.; writing—original draft preparation, H.K. and S.Y.Y.; writing—review and editing, H.K.; visualization, H.K. and C.O.; supervision, H.K.; project administration, H.K.; funding acquisition, H.K. All authors have read and agreed to the published version of the manuscript.

**Funding:** This research received no external funding.

**Institutional Review Board Statement:** The study was conducted in accordance with the Declaration of Helsinki, and approved by the Institutional Review Board of Wonju Severance Christian Hospital (CR321137 and date of approval 12 July 2023).

**Informed Consent Statement:** Not applicable.

**Data Availability Statement:** Dataset available on request from the authors.

**Conflicts of Interest:** The authors declare no conflicts of interest.

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
