# Peer review of "Intravoxel Incoherent Motion Diffusion-Weighted MRI, Fat Quantification, and Electromyography: Correlation in Polymyositis and Dermatomyositis"

_tomography, doi:10.3390/tomography10030029_

Round 1

Reviewer 1 Report

Comments and Suggestions for Authors

This is a well-designed study to evaluate for correlations between IVIM and fat-fractionation values on MRI and findings on EMG in patients with dermatomyositis and polymyositis.  

The Methods are exlpained well, and the MRI parameters and image analysis are clearly defined. The statistical methods are valid. The figures show the method for the MRI analysis and the results very well. Table 4 is titled incorrectly as "Results of interobserver agreement analysis” when it shows the relationship between MRI values and spontaneous pathologic activity. That should be corrected.  

The first paragraph of the Discussion is the instructions to authors, and should be removed. The Discussion otherwise nicely summarizes the findings and places them in the context of prior research. The References section is thorough.  

Tha major weaknesses of the study are the small sample size, the evaluation of only 2 types of inflammatory myopathy, and the limited anatomic coverage provided by placing th ROI on only one slice. The authors acknowledge these limitations in the discussion.  

In summary, this is a good study and manuscript. I recommend accepting it after correction of the 2 minor errors described above.

Author Response

Q1. Table 4 is titled incorrectly as "Results of interobserver agreement analysis” when it shows the relationship between MRI values and spontaneous pathologic activity. That should be corrected.

A1. Thank you for your comments. We have changed the title of table 4 to "Analysis of IVIM-Derived Parameters, Fat Fraction, and ADC Value".

Q2. The first paragraph of the Discussion is the instructions to authors and should be removed. 

A2. Thank you for your comments. We have removed the paragraph.

Reviewer 2 Report

Comments and Suggestions for Authors

This manuscript is a clear and succinct description of the application of two quantitative MRI methods in the evaluation of skeletal muscle inflammation. Although a relatively small cohort, the study is well-described, and the results show the comparison of several diffusion (IVIM) and fat quantification biomarkers with invasive electromyography results.

There are two results which I believe the authors should discuss in further detail:

1) It is not clear which and how many of the study participants had pathologic spontaneous activity. This should be added to Table 1 or as a separate table.

2) The authors report a significant correlation in fat(%) with both amplitude and duration, however there appears to be a large spread in fat(%) at low amplitude and low-mid duration. Could the authors comment on what appears to be two groups (low and high fat(%)) and how these relate to the patients that showed pathologic spontaneous activity? Could it be related to the inclusion of patients with relatively early-stage inflammatory myopathies? Some discussion is needed,

The remaining minor points should be consider/amended:

Line 26 – f values are percentages, add %.

Lines 57 and 98 – suggest replacing “contrast material” with “contrast agent”.

Line 131 – radiologists had five years of experience each or together? Clarify.

Line 189 – Table 4 caption incorrect.

Table 4 – Consider stating D*, D and ADC in × 10-3 mm2/s, as is common.

Lines 200-202 – Remove first paragraph of discussion. Explanatory text.

Line 226 – Add space between [19] and A.

Line 234 – remove “disturbs”.

Line 282 – missing reference information.

Lines 277 to 320 – some journal names incorrectly capitalised. Amend.

Comments on the Quality of English Language

The English is clear.

Author Response

Q1. It is not clear which and how many of the study participants had pathologic spontaneous activity. This should be added to Table 1 or as a separate table.

A1. Thank you for your comments. We have added information on study participants showing pathologic spontaneous activity to Table 1.

Q2. Could the authors comment on what appears to be two groups (low and high fat(%)) and how these relate to the patients that showed pathologic spontaneous activity? Could it be related to the inclusion of patients with relatively early-stage inflammatory myopathies?

A2. Thank you for your comments. We have added these sentences to the discussion.

In our study, two of the participants showed a fat fraction of more than 20% in the affected muscles, whereas the remaining ten patients exhibited a fat fraction of less than 7%. Due to the small number of patients with a high fat fraction, the results in terms of duration and amplitude on their EMG could appear more pronounced comparatively. These patients had disease durations of 10 and 24 months, respectively, which are relatively longer compared to other patients. This suggests that future studies may need to include a larger group of patients with a variety of disease durations and higher fat fractions for further validation. All patients with a fat fraction of more than 20% showed pathologic spontaneous activities on EMG. The fat fraction serves as a quantitative indicator of fatty cell infiltration in the affected muscle. Our study confirmed the presence of pathologic spontaneous activities on EMG, which can result from severe fatty cell infiltration due to chronic inflammation, leading to the loss and disorganization of muscle fibers.

Q3. The remaining minor points should be consider/amended

A3. Thank you for your comments. We have made the corrections to your points.

Reviewer 3 Report

Comments and Suggestions for Authors

This manuscript provides a good understanding of the relationship between MRI-based IVIM-DWI, quantification of fat and electromyography (EMG) in patients with polymyositis and dermatomyositis. The article is well-designed, and the technique is well-described and reproducible. The table with the acquisition protocol is commendable.

Discussion
Lines 200-203 seem to be instructions for the authors.

I would suggest adding how this technique could be useful in the diagnostic and treatment path of patients with rheumatic disease.

After studying again the manuscript I would like to add the following comment:

A control group of healthy individuals could strengthen the study by highlighting the statistical deviation of MR signal in DWI and quantification of fat between healthy muscles and DM and PM patients.

Author Response

Q1. Discussion Lines 200-203 seem to be instructions for the authors.

A1. Thank you for your comments. We have removed the paragraph.

Q2. I would suggest adding how this technique could be useful in the diagnostic and treatment path of patients with rheumatic disease.

A1. Thank you for your comments. We have added these sentences to the discussion.

Our study is the first to compare IVIM-DWI parameters and fat fraction, which are quantitative MRI measures, with EMG findings in inflammatory myopathy. EMG is an important tool for predicting degeneration and pathologic changes in affected muscles in the diagnosis of inflammatory myopathy. However, EMG is an invasive diagnostic tool and cannot be used if the patient's condition is unstable. Predicting EMG findings using MRI could offer an alternative for patients where EMG cannot be performed. Moreover, quantitative analysis using MRI can enhance the understanding of the pathophysiology of inflammatory myopathy and enable monitoring of the disease's treatment progress.